# Wintertime hygroscopicity and volatility of ambient urban aerosol particles

Joonas Enroth[1], Jyri Mikkilä[1], Zoltán Németh[2], Markku Kulmala[1], and Imre Salma[2]

[1] Department of Physics, P.O. Box 64, 00014, University of Helsinki, Helsinki, Finland
[2] Institute of Chemistry, Eötvös University, P.O. Box 32, H–1518, Budapest, Hungary

*Correspondence to*: Imre Salma (salma@chem.elte.hu)

**Abstract.** Hygroscopic and volatile properties of atmospheric aerosol particles with dry diameters of (20,) 50, 75, 110 and 145 nm were determined in situ by using a VH-TDMA system with a relative humidity of 90% and denuding temperature of 270 °C in central Budapest during two months in winter 2014–2015. The probability density function of the hygroscopic growth factor (HGF) showed a distinct bimodal distribution. One of the modes was characterised by an overall mean HGF of approximately 1.07 (this corresponds to a hygroscopicity parameter κ of 0.033) independently of the particle size, and was assigned to nearly hydrophobic (NH) particles. Its mean particle number fraction was large, and it was decreasing monotonically from 69 to 41% with particle diameter. The other mode showed a mean HGF increasing slightly from 1.31 to 1.38 (κ values from 0.186 to 0.196) with particle diameter, and it was attributed to less hygroscopic (LH) particles. The mode with more hygroscopic particles was not identified. The probability density function of the volatility GF (VGF) also exhibited a distinct bimodal distribution with an overall mean VGF of approximately 0.96 independently of the particle size, and with another mean VGF increasing from 0.49 to 0.55 with particle diameter. The two modes were associated with less volatile (LV) and volatile (V) particles. The mean particle number fraction for the LV mode was decreasing from 34 to 21% with particle diameter. The bimodal distributions indicated that the urban atmospheric aerosol contained an external mixture of particles with a diverse chemical composition. Particles corresponding to the NH and LV modes were assigned mainly to freshly emitted combustion particles, more specifically to vehicle emissions consisting of large mass fractions of soot likely coated with or containing some water-insoluble organic compounds such as non-hygroscopic hydrocarbon-like organics. The hygroscopic particles were ordinarily volatile. They could be composed of moderately transformed aged combustion particles consisting of partly oxygenated organics, inorganic salts and soot. The larger particles contained internally mixed non-volatile chemical species as a refractory residual in 20–25% of the aerosol material (by volume).

## 1 Introduction

Aerosols influence our life and environment in multiple ways. They affect the climate system and water cycling (Boucher et al., 2013), ecosystems (Mercado et al., 2009), human health and welfare (Lelieveld et al., 2015), built environment (Brimblecombe, 2016) and visibility (Davidson et al., 2005). One of the key factors in all these processes and their consequences is the size of particles. The size distribution of atmospheric aerosol particles is primarily determined by their formation process. It can be, however, influenced further in the air by condensation or evaporation of vapours (McMurry and Stolzenburg, 1989). Water vapour is the most abundant vapour in the troposphere; it is a minor constituent of the air, while the other vapours or their precursors are present in trace concentrations. Water uptake by aerosol particles or water evaporation from particles under subsaturated conditions (thus its hygroscopicity) are explained by Köhler theory. The interactions between water vapour and particles are affected by the size, chemical composition, surface tension and water activity of particles on one side and the meteorological conditions, specifically air temperature ($T$) and relative humidity (RH) on the other side. These relationships have a special climate and environmental relevance for droplet formation on cloud condensation nuclei (CCN). Some particles from specific emission or formation sources contain relatively large amounts of semi-volatile chemical compounds (other than water), which can evaporate from the particles depending on ambient conditions, mainly on air $T$

according to their partitioning between the condensed and gas phases. It is worth noting that dissolution of some atmospheric gases into droplets and their subsequent water-phase chemical reactions could also change the particle size (Meng and Seinfeld, 1994; Kerminen and Wexler, 1995). The dissolution takes place preferentially with accumulation-mode particles because of the largest total surface area of this mode (and causes the splitting the accumulation mode into the condensation and droplet submodes). Nevertheless, the related absolute increase in the diameter is small due to the fact that the relevant gases are ordinarily present in trace concentrations, and the relative diameter increments are modest considering the diameter range of these particles. Influence of the dissolution-chemical reaction process on the particle size is smaller than that of condensation of water vapour or evaporation of semi-volatile chemical species including water, and therefore, it can be usually disregarded. The hygroscopic and volatile properties have been increasingly recognised and used for global modelling purposes (McFiggans et al., 2006; Pringle et al., 2010; Rissler et al., 2010). At the same time, determination of the changes in particle size for different dry diameters under various RHs and $T$s can also supply valuable indirect in situ information on the chemical composition, extent of the external or internal mixing, surface coatings and chemical reactivity of particles (Massoli et al., 2010; Wu et al., 2016; Cai et al., 2017). The hygroscopic properties under subsaturated conditions and volatility of ambient particles can be studied by on-line Tandem Differential Mobility Analyzer (TDMA) systems, including the Volatility-Hygroscopicity TDMA (VH-TDMA). The method has particular importance since it is rather difficult to obtain direct information on the chemical composition of ambient ultrafine (UF) particles (with $d < 100$ nm) due to the small total mass represented by them, and their dynamic processes. The measurements can also be used for determining the time scales for atmospheric chemical or physicochemical transformations of freshly emitted nearly hydrophobic particles into more hygroscopic types of particles, which is a relevant issue in particular in cities. The technique has been successfully applied in remote, marine, rural or semi-urban atmospheric environments worldwide, and the results and conclusions were summarised in a review paper by Swietlicki et al. (2008). Measurements on complex ambient mixtures explicitly in urban centres have also been increasing (Cocker et al., 2001; Baltensperger et al., 2002; Ferron et al., 2005; Massling et al., 2005; Tiitta et al., 2010; Juranyi et al., 2013; Kamilli et al., 2014; Wu et al., 2016; Cheung et al., 2016; Cai et al., 2017). It turned out from the studies that the hygroscopic and volatile properties vary strongly with the location and origin of air masses, and that the urban-type air pollution can strongly influence them. Moreover, multicomponent chemical mixtures of inorganic salts and organic compounds including coatings, which are typically present in ambient aerosol particles in cities, are poorly characterised and understood. The potential effects concern many people since there is a spatial coincidence between the air pollution and population density in cities.

As part of a long-term cooperation between the Eötvös University, Budapest and University of Helsinki in the field of atmospheric aerosol research (e.g. Salma et al., 2016b), in-situ VH-TDMA measurements were performed in central Budapest for the first time together with other supporting measurements for two months in winter. The major objectives of the present paper are to present the results on hygroscopic and volatile properties of aerosol particles with various dry diameters, to improve the geographical coverage with this type of the measurements with regard to the Carpathian Basin, to interpret the two data sets jointly, to identify and discuss the mixing state and source processes of particles, and to conclude their implications on urban aerosol. The results and conclusion are to contribute to the improved understanding of hygroscopic and volatile properties of the urban-type atmospheric environment in general.

## 2 Experimental methods

The measurements took place at the Budapest Platform for Aerosol Research and Training (BpART) research facility (Salma et al., 2016a) in Budapest from 09–12–2014 to 09–02–2015. The location represents a well-mixed, average atmospheric environment for the city centre. The sampling inlets were set up at heights between 12 and 13 m above the street level of the

closest road and were located in a distance of 85 m from the river Danube. Distance of Budapest to the Adriatic Sea, Baltic Sea, Black Sea and North Sea are approximately 450, 780, 830 and 1200 km, respectively. The mean and standard deviation (SD) of air $T$ and RH inside the BpART during the measurement campaign were 20±1 °C and 28±8%, respectively, and the air $T$ stratification was avoided by extra fans located properly inside the facility. The wintertime median concentrations of elemental carbon (EC), organic carbon (OC) and particulate matter (PM) mass in the $PM_{2.5}$ size fraction were 0.97, 4.9 and 25 $\mu g\ m^{-3}$, respectively (Salma et al., 2017). The mean contributions of EC and organic matter (OM, with an OM/OC mass conversion factor of 1.6) to the $PM_{2.5}$ mass and SDs were 4.8±2.1% and 37±10%, respectively, while the contribution of $(NH_4)_2SO_4$ and $NH_4NO_3$ derived in an earlier study in spring were 24% and 3%, respectively. The on-line instruments deployed in the present campaign were a VH-TDMA system, a differential mobility particle sizer (DMPS) system and various meteorological sensors, which were operated in parallel.

The key instrument related to this study was a VH-TDMA system described by Hakala et al. (2017). In short, the sample air with a flow rate of 2 L $min^{-1}$ was dried to RH<10% using a silica-gel diffusion dryer at indoor temperatures, then the equilibrium electric charge distribution of the aerosol particles was ensured by using a C-14 radioactive bipolar charger. Particles with median dry diameters of 20, 50, 75, 110 and 145 nm were preselected in a narrow quasi-monodisperse size range by a 10.9-cm long Vienna-type differential mobility analyser (DMA1). The dry diameters were selected considering the shape of the particle number size distribution in Budapest to cover the Aitken and accumulation modes. The inlet air flow was then separated into two flows with equal rates, which continued to either a humidifying chamber with an enhanced RH to allow water uptake, or to a thermal denuder with a higher $T$ to allow evaporation. The humidifier was a Gore-Tex tube with a length of 2.5 cm in a heated MilliQ water bath. The RH in the humidifier was set to 85% in the very beginning of the campaign, and it was increased to 90%, which is commonly regarded as the standard humidification. The mean RH and its standard deviation (SD) were 90.0±0.4% during the second (much longer) part of the campaign. The RH was maintained by a PID controller with a relatively slow response, which ensured that the RH during each scan remained stable. The mean difference in the RH values between scans remained <0.16%. The thermal denuder was a well-insulated cylindrical metal tube maintained at a mean temperature and SD of 270.0±0.1 °C by electrical resistance wires and heat insulations. This temperature was selected as a compromise by considering the thermal behaviour of relevant and abundant inorganic salts, several types of organic compounds and soot (Park et al., 2009; Hakala et al., 2017) and the conclusions of varying temperature protocols in previous studies with real ambient aerosol particles (Tritscher et al., 2011; Hong et al., 2014; Cai et al., 2017). Atmospheric sulphates, nitrates and most organics are usually volatile at this temperature, while soot (and some organic polymers) remain refractory. The two main classes can be advantageously combined with the hygroscopicity measurements. The centreline residence times of particles in the humidifying section and thermal denuder were 0.6 and 0.5 s, respectively, which is adequate for equilibrium particle growth/evaporation of typical inorganic salts (Chan and Chan, 2005). The humidifier and denuder were operated in parallel, giving simultaneous data on hygroscopic and volatile properties of particles. The advantage of this combination is a better time resolution, which is an important factor for relatively rapidly varying urban atmospheric environments. The changes in particle diameter after the humidifier were determined by a Vienna-type DMA (DMA2a) with a length of 28 cm and a condensation particle counter (CPC, TSI, model 3772, operated with an option of water removal on). The sheath air flow of the DMA2a was also humidified to the selected RH. The changes in particle diameter after the thermal denuder were measured by a Vienna-type DMA (DMA2b) with a length of 10.9 cm and a CPC (TSI, model 3010). The concentration of the size-selected particles in the denuded air flow was low enough to prevent recondensation after the denuder (Park et al., 2009). The $T$ and RH of the air flows were monitored just at the entrance to the second set of DMAs. The RH was measured by using a chilled-mirror dew point hygrometer located in the excess air flow of the DMA2a. The DMA1, DMA2a and DMA2b were all operated with a closed loop sheath air flow system with flow rates of 20, 10 and 9 L $min^{-1}$, respectively. The time resolution of the VH-TDMA system was 18 min, which has high importance in dynamically changing urban atmospheres. Calibration,

quality check of the hygroscopic GF and operation check of the volatility GF for all selected diameters was performed by nebulising dilute solution of analytical grade $(NH_4)_2SO_4$ into the filtered inlet air flow of the VH-TDMA system. The system was operated according to international recommendations (Duplissy et al., 2009; Massling et al., 2011; Cheung et al., 2016; Cai et al., 2017).

The DMPS used was a flow-switching type system (Salma et al., 2011b; Salma et al., 2016a). Its main components are a radioactive (Ni-60) bipolar charger, a Nafion semi-permeable membrane dryer, a 28-cm long Vienna-type DMA and a butanol-based CPC (TSI, model 3775). It measures particle number size distributions in an electrical mobility diameter range from 6 to 1000 nm in the dry state of particles (with a RH<30%) in 30 channels with a time resolution of approximately 8 min. The

sample flow rate is 2.0 L min$^{-1}$ in high-flow mode, and 0.30 L min$^{-1}$ in low-flow mode with sheath air flow rates 10 times larger than for the sample flows. The DMPS measurements were performed according to the recommendations of the international technical standard (Wiedensohler et al., 2012). Weather shield and insect net were adopted to both aerosol inlets. The meteorological data were available from Urban Climatological Station of the Hungarian Meteorological Service operated at a height of 10 m above the roof level of the building (at a height of 39 m above the ground) in a distance of about 40 m from

the BpART facility, and from a simpler on-site meteorological station. Standardised meteorological measurements of $T$, RH, wind speed (WS) and wind direction (WD) were recorded with a time resolution of 10 min.

## 3 Data evaluation

The ratio of the particle diameter after the treatment in the humidifier or thermal denuder to the initially selected diameter is the diameter growth factor (HGF and VGF, respectively). It expresses both possible growth and shrinkage of particles. For

ambient particles, the HGF and VGFs are spread in a distribution, which was measured by holding the particle size selected by the DMA1 constant, and scanning by the second set of DMAs (DMA2a and 2b) through possible diameter ranges that correspond to HGFs from 0.9 to 2.0, and to VGFs from 1.3 to 0.3. This resulted in a distribution of the HGF or VGF, which is referred as growth factor probability density function (PDF). It is, however, a smoothed and skewed form, which needs to be mathematically inverted to derive the distribution accurately since the transfer function of the second set of DMAs has a finite

width, and since their total transfer probabilities depend on the classifying voltage and operational parameters of the DMA (Swietlicki et al., 2008; Gysel et al., 2009). The inversion process yields the true growth factor PDF for each scan, and it is also normalised to unity. The inversion was accomplished by applying the program package TDMAinv (Gysel et al., 2009) in IGOR Pro software. The program package was shown to be robust, appropriate and adequate in several previous studies (e.g. Liu et al., 2011; Hong et al., 2014). The hygroscopic property was also expressed by hygroscopicity parameter (κ value, Petters

and Kreidenweis, 2007). The κ value is a compound parameter with more sophisticated interpretation than the hygroscopic GF, but it can be used in the Köhler model as proxy for the chemical composition without explicitly knowing the density, molecular mass, van't Hoff factor and osmotic coefficient or dissociation number of each chemical component. It was calculated for each HGF as:

$$\kappa = (\mathrm{HGF}^3 - 1)\left[\frac{1}{S}\exp\left(\frac{4\sigma M_w}{RT\rho_w D_d \mathrm{HGF}}\right) - 1\right], \tag{1}$$

where $S$ is the saturation ratio of water ($S$=RH, when RH is expressed as a fraction), σ is the surface tension of the droplet-air interface at the composition of the droplet (see later), $M_w$ is the molecular mass of water (0.018015 kg mol$^{-1}$), $R$ is the universal gas constant (8.3145 J mol$^{-1}$ K$^{-1}$), $T$ is the droplet temperature (obtained directly from the DMA as 298.15 K), $\rho_w$ is the density of water in the droplet (997.05 kg m$^{-3}$ at 298.15 K), and $D_d$ is the dry diameter of the particle. The σ was assumed to be that of pure water (σ=72.0 mN m$^{-1}$ at 298.15 K). It was shown that some organic chemical species in atmospheric aerosol particles

such as humic-like substances (HULIS) are surface active and can lower the surface tension of the water droplet (Facchini et al., 1999; Salma et al., 2006). The use of a smaller σ than for pure water yields smaller κ values. The depression of σ in time is mainly controlled by diffusion of HULIS from the bulk of the droplet to its surface, and it takes several hours to reach the thermodynamic equilibrium at medium concentrations (Salma et al., 2006). This also means that the extent of the actual decrease is kinetically limited to larger than equilibrium values, and therefore, the utilisation of σ=60 mN m$^{-1}$ is expected to represent a lower estimate. The κ values obtained by this surface tension changed by a mean factor of 0.97–0.98 for particles with a diameter of 70, 110 and 145 nm, while the mean factor for particles with a diameter of 50 nm became 0.95. This all implies that the possible alterations related to the lower surface tension than that of water are small with respect to estimated experimental uncertainties in the determination of the HGFs (up to 15–20%), and that the calculations with σ for water seem to be a sensible approach to reality in the size range and RH considered in the present study. This is also confirmed in an earlier article, according to which the sensitivity of hygroscopic growth to surface tension becomes more important with decreasing dry particle diameter and increasing RH since these are the conditions under which HGFs or κ values are most sensitive to the Kelvin factor (Swietlicki et al., 2008). Therefore, the surface tension of pure water was adopted in the calculations. The soluble particle volume fraction is often used to classify different groups of hygroscopic growth in order to facilitate data comparability. Here, we utilise κ values and HGFs to define commonly observed hygroscopic groups (Liu et al., 2011). The limits of the groups are, however, not exactly determined. They are often set as 1) κ≤0.10 (HGF≤1.21): nearly hydrophobic (NH) particles; 2) κ≈0.10–0.20 (HGF=1.21–1.37): less hygroscopic (LH) particles, and 3) κ>0.20 (HGF>1.37): more hygroscopic (MH) particles. The HGFs above refer to particles with a diameter of 100 nm and RH=90%, and vary with $D_d$ and RH. On the one side, small values of κ indicate low CCN-active behaviour. As κ approaches 0, the particle resemble an insoluble but wettable particle. On the other side, the upper limit for the most hygroscopic species typically found in the marine atmospheric aerosol particles (e.g. NaCl) is κ≈1.4 (HGF≈1.85) at RH=90% (Petters and Kreidenweis, 2007).

The volume fraction remaining (VFR) after the thermal treatment was calculated from the VGF assuming spherical shape of particles both before and after the treatment (Häkkinen et al., 2012). The mean growth factor was calculated using the retrieved GF-PDF as:

$$\overline{\mathrm{GF}} = \frac{\sum_i \mathrm{GF\text{-}PDF}_i \times \mathrm{GF}_i}{\sum_i \mathrm{GF\text{-}PDF}_i}, \tag{2}$$

where GF-PDF$_i$ is the value of the PDF at GF$_i$. Number fraction (NF) of particles in the different modes was computed as:

$$\mathrm{NF} = \frac{\sum_i^j \mathrm{GF\text{-}PDF}_i}{\sum_i \mathrm{GF\text{-}PDF}_i}. \tag{3}$$

The partial sum in the numerator of Equation 3 was either calculated 1) for the HGF interval from $i$=0.9 to $j$=1.2 for the NH mode, and from $i$=1.2 to $j$=2.3 for the LH mode (see Fig. 2a), or 2) for the VGF interval from $i$=0.3 to $j$=0.8 for the V mode, and from $i$=0.8 to $j$=1.3 for the LV mode (see Fig. 2b). Complete evaporation of particles in the thermal denuder and the possible size-dependent particle loss within the instrument were not taken into account in the present study. The fraction of completely evaporated particles was, however, roughly estimated to be between 60 and 32% for particle diameters of 50 and 145 nm, respectively. Evaluation of the measured DMPS data was performed according to the procedure protocol recommended by Kulmala et al. (2012). Mathematically inverted DMPS data were utilised for calculating particle number concentrations in the particle diameter ranges from 6 to 100 nm (UF particle number concentration, $N_{<100}$), from 100 to 1000 nm ($N_{>100}$) and from 6 to 1000 nm (total particle number concentration, $N$). Median particle number size distribution was derived from the individual concentration data. Diurnal variation for a selected variable was obtained by averaging the measured data for the corresponding hour, and then by averaging the hourly means for the whole time interval.

## 4 Results and discussion

Most of the measurement period consisted of mild and moist winter weather. The air $T$ and RH ranged from –9 to 17 °C, and from 40 to 99%, respectively with means and SDs of 3.0±4.2 °C and 84±13%, respectively. No extreme wind conditions were encounter; the mean WS and its SD were 1.9±1.7 m s$^{-1}$. The median $N$ and UF particle number concentration were $5.0 \times 10^3$ cm$^{-3}$ and $4.1 \times 10^3$ cm$^{-3}$, respectively for the measurement campaign, and the mean UF/$N$ concentration ratio and its SD were 82±9%. The median concentrations are below the annual medians (Salma et al., 2016b). There were four quantifiable new aerosol particle formation and growth events identified in the DMPS data set during the VH-TDMA measurement campaign in winter. Unfortunately, only one of them was fully covered by the VH-TDMA system, and therefore, no representative conclusion can be drawn for them (cf. Kerminen et al., 2012; Wu et al., 2016).

### 4.1 Growth factor spectra and their temporal variability

Time series of the HGF-PDF for particles with a dry diameter of 145 nm at RH=90%, and particle number concentrations in various size fractions for approximately 1 week are shown in Fig. 1 as example. The white bands indicate missing data. The HGF-PDF exhibited bimodal distribution. The mode with the smaller HGF values represented NH particles, while the other mode with the larger HGFs was related to LH particles. The two modes were constantly present and were well separated during the whole campaign despite the large variations in $N$ and $N_{>100}$ concentrations. Fig. 1 also confirms that the classification of the NH and LH groups (see Sect. 3) in Budapest was appropriate. The contribution of the two modes was size dependent (which is quantified in Table 1 and is discussed below). Importance of the NH mode decreased with particle size. The smallest, 20-nm particles already resembled a unimodal distribution with only NH mode for most of the time. Two distinct modes showed up in the VGF-PDF spectrum as well, and they were also constantly present during the whole campaign. The mode with the VGF>0.8 was assigned to less volatile (LV) particles, while the mode exhibiting VGF≤0.8 was associated with volatile (V) particles. These 2 groups of particles (2 modes) based on volatile properties appear satisfactory under actual experimental and atmospheric conditions and for our present purposes. Other, more detailed classifications are also possible (Cheung et al., 2016). The bimodal distributions indicated that the particles had separate chemical composition, thus they were present in external mixture, except for 20-nm particles, which basically showed internal mixture of constituents. The relative contribution of the modes were highly variable, which can be explained by the temporal variability in the intensity of the major emission and formation sources of the corresponding particles, in their atmospheric transformation processes, and by the influence of different air masses.

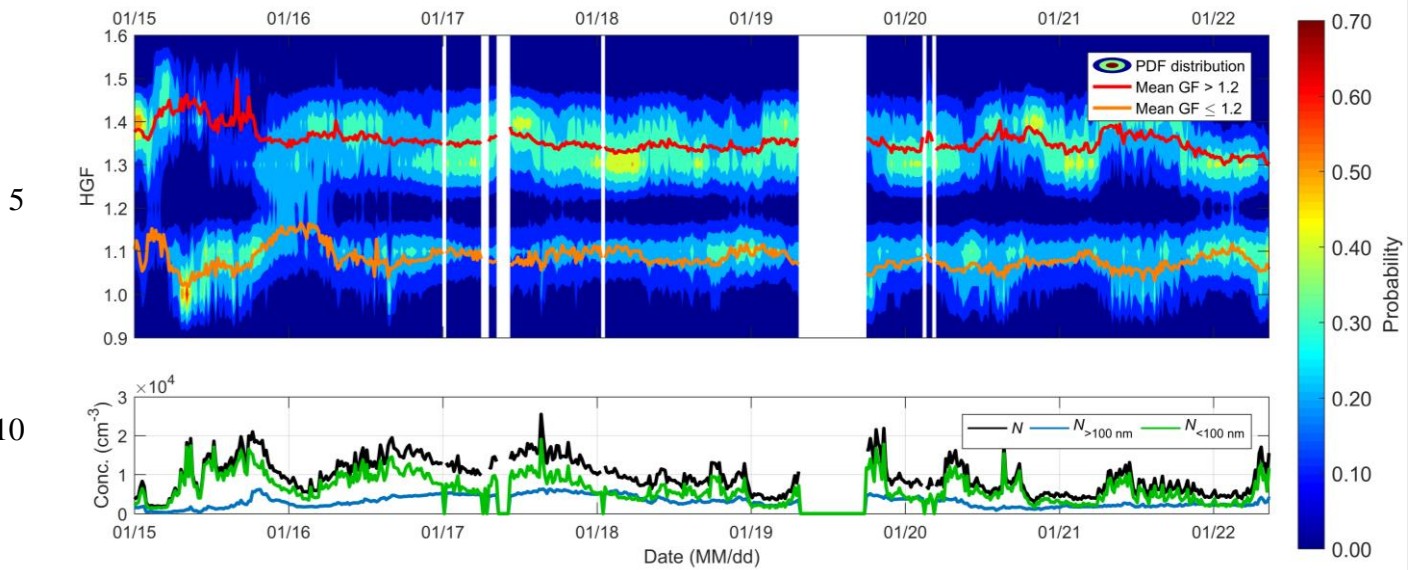

**Figure 1.** Temporal variability of the hygroscopic diameter growth factor probability density function (HGF-PDF) (upper panel) at a RH=90% for particles with a dry diameter of 145 nm from 15 to 22 January 2015. The orange line displays the mean HGF of the nearly hydrophobic mode (HGF$_{NH}$), while the red line shows the mean HGF of the less hygroscopic mode (HGF$_{LH}$). Total particle number concentration (*N*, black line), and concentrations of $N_{<100}$ (green line) and $N_{>100}$ (blue line) are also shown on the lower panel.

## 4.2 Averages of hygroscopic and volatile properties

Median particle number size distribution together with the mean HGF-PDF and the mean VGF-PDF for different particle diameters are shown in Fig. 2. The size distribution (Fig. 2a) demonstrates that particles in cities mainly originate from multiple, complex local and regional emission and formation sources. Its broadening was caused by the averaging the individual data as well. Figure 2a also shows that the selected diameters of 20, 50, 75, 110 and 145 nm represent the plateau of the size distribution, and that the median ambient concentrations of these particles were similar to each other (approximately $1.2 \times 10^3$ cm$^{-3}$). The particles with a diameter of 20 nm were omitted from the volatility evaluation and Fig. 2c because they appeared at the lower end of the VGF range after the shrinkage, thus their diameters were close to the detection limit of the CPC (TSI model 3010) used as the detector, and they were also subjected to enhanced diffusion losses. The mean HGF and VGF clearly exhibited bimodal character. The relative contribution of NH mode in the HGF-PDF (Fig. 2b) was decreasing with particle diameter, while there was no similar obvious tendency for the VGF-PDF (Fig. 2c).

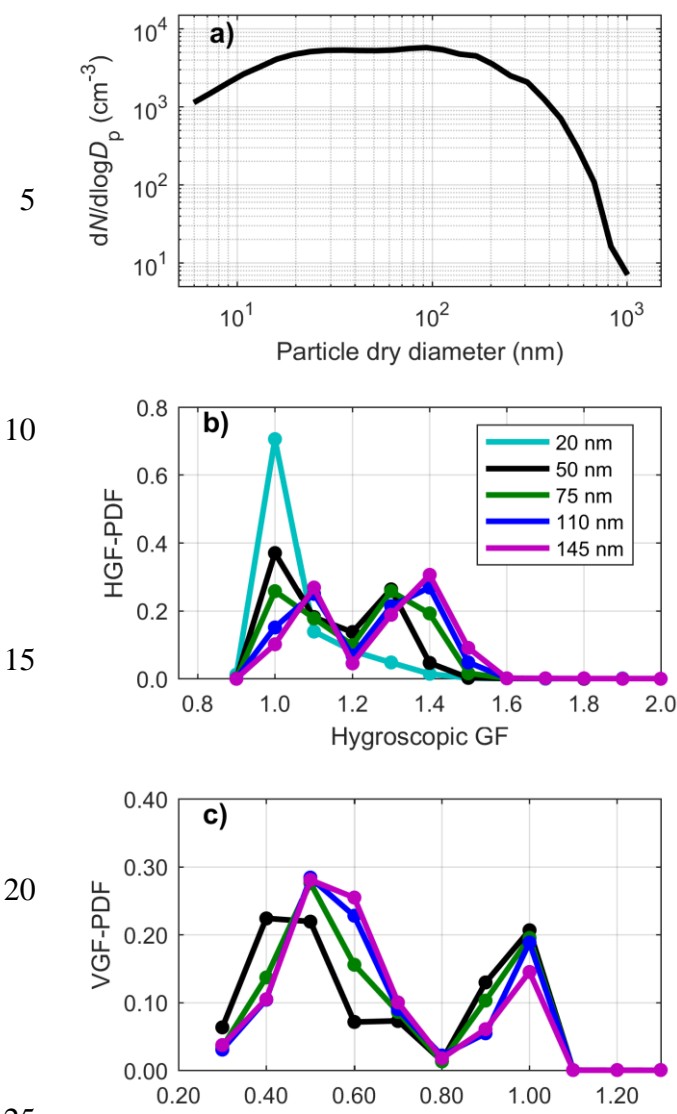

**Figure 2.** Median particle number size distribution for the whole campaign (panel a) together with the mean hygroscopic diameter growth factor probability density function (HGF-PDF; panel b) and volatility diameter growth factor probability density function (VGF-PDF; panel c) for different dry particle diameters. The panels b and c have shared legends.

The mean HGFs and κ values separately for the NH and LH modes, and the mean VGFs separately for the LV and V modes together with the number fractions of particles in each mode for the different dry diameters are summarised in Table 1. Relatively small size dependency was observed for the growth of the NH and LH modes as well as for the shrinkage of the LV and V modes, while the number fraction of particles the modes varied substantially. Specifically, the mean $HGF_{NH}$ mode of approximately 1.08 did not seem to depend on particle size, while the mean $NF_{NH}$ decreased monotonically and substantially, from 69 to 41% with particle diameter. It is mentioned that approximately 85% of particles with a diameter of 20 nm were associated with the NH mode, and they showed the largest time variation. The variation in the $HGF_{NH}$ mentioned above was small. The differences in the growth behaviour are expected to become larger with RH under sub-saturated conditions, so a more reliable interpretation of this change (variability or tendency) would likely be achieved by dedicated experiments with RHs of 95% or even higher. The situation was similar for the LV particles, thus the mean $VGF_{LV}$ stayed constant on a level of approximately 0.96 independently of the particle size, while the particle number fraction in the LV mode ($NF_{LV}$) was decreasing from 34 to 21%, although its extent was less pronounced than for the $NF_{NH}$ particles. The NH mode and the LV mode were jointly related to freshly emitted combustion particles consisting of large mass fractions of soot and some water-insoluble organic compounds (Liu et al., 2013; Cheung et al., 2016). The latter species can be a mixture of non-hygroscopic hydrocarbon-like organic aerosol in the condensed phase, and adsorbed Volatile Organic Compounds (VOCs) from the gas

phase in relatively large amounts. The tendencies in the hygroscopic and volatile properties, and in the particle number fraction for the modes mentioned are ordinarily observed in urban VH-TDMA studies (e.g., Ferron et al., 2005) since the particle number size distribution of soot from traffic emissions peaks between 50 and 100 nm (Weingartner et al., 1997). It was previously concluded that uncoated fresh soot particles (although of somewhat larger diameters than 145 nm) showed neither hygroscopic growth nor water activation, while their coating with succinic acid, sulphuric acid or polyaromatic hydrocarbons (PAHs) influenced the hygroscopic growth in a complex way (Henning et al., 2012). Polyaromatic hydrocarbons are usually not hygroscopic; they are slightly soluble or even insoluble in water, and their solubility decreases with molecular mass. Sulfuric acid reacted with the PAHs and likely formed products with lower molecular mass than the initial PAHs. These products had a higher solubility in water, and as a consequence, the hygroscopic growth and activated fraction increased due to 1) the products of this reaction and 2) the unconsumed coating fraction itself. The diameter change depended on the amount and type of the coatings, on the "humidity history" of particles (coating by solution layer or solid film) and on the carrier gas used in the experiments. The interaction between the soot particles and water vapour also included hygroscopic shrinkage. Hygroscopic GFs up to 1.11 were obtained with succinic acid at RH=98% for particles with 375 and 500 nm dry diameter (Henning et al., 2012). It has to be noted that succinic acid coating on soot particles appears to be a good example for oxygenated organic substances in atmospheric environments strongly influenced by biogenic activities, while the situation with the coatings in urban environments could be different. Atmospheric fresh soot agglomerates have fractal structure, which can be reconstructed when exposed to high RH due to capillary condensation or due to filling up cavities leading to compaction (Weingartner et al., 1995). The particles after the structural change (collapse) become less fractal-like or more compact, which results in decreased electrical mobility diameter, and thus leads to particle shrinkage (HGF<1), and finally, to underestimated HGF if obtained by diameter-based methods. The restructuring has been observed for different soot types (Weingartner et al., 1995; Tritscher et al., 2011) and for particles with a $D_d$ >100 nm (Martin et al., 2012). The $HGF_{LH}$ increased monotonically but in a modest way (from 1.31 to 1.38) with particle size, while the $NF_{LH}$ increased with diameter from 29% (for $D_d$=20 nm) to 59%. It is worth realising that the HGFs for pure $(NH_4)_2SO_4$ and $NH_4NO_3$ (Park et al., 2009) are substantially larger than the measured values. These particles can be composed of moderately transformed aged soot-containing combustion particles comprising also partly oxygenated organics and inorganic salts (Duplissy et al., 2011; Liu et al., 2013). This view is further confirmed by a high-resolution transmission electron microscopy with electron energy-loss spectroscopy (TEM/EELS) study of individual particles at the BpART facility (Németh et al., 2015). Both regional background sources and urban (local) emissions can contribute to these particles (Swietlicki et al., 2008), which results in complex mixtures. The whole explanation is also coherent with the fact that the mixing state of soot particles is mostly determined close to their emission sources (Liu et al., 2013). Mean number fraction of volatile particles ($NF_V$) was increasing from 66 to 79% with particle diameter, and it was accompanied by almost identical mean VGFs (0.49–0.54 with a relative SD of approximately 9%). All these correspond to earlier similar observations in urban environments or metropolitan regions, and support the idea that particles with larger volatility consist of soot particles internally mixed (coated) with volatile material (Cheng et al., 2006; Wehner et al., 2009; Cheung et al., 2016).

**Table 1.** Mean hygroscopic growth factor (HGF) and mean hygroscopicity parameter ($\kappa$) separately for nearly hydrophobic (NH) mode (HGF<1.2) and less-hygroscopic (LH) mode (1.45>HGF≥1.2) obtained for different particle diameters ($D_d$) at a mean RH of 90%, and mean volatility growth factor (VGFs) separately for less volatile (LV) mode and for volatile (V) mode for different $D_d$ values at a mean denuding temperature of 270 °C. Mean number fraction (NF, in %) with respect to the total particle number concentration, and SD for each mode are also given.

| $D_d$ | 50 nm | | 75 nm | | 110 nm | | 145 nm | |
|---|---|---|---|---|---|---|---|---|
| Property | Mean | SD | Mean | SD | Mean | SD | Mean | SD |
| $HGF_{NH}$ | 1.07 | 0.04 | 1.07 | 0.04 | 1.09 | 0.03 | 1.09 | 0.03 |
| $\kappa_{NH}$ | 0.034 | 0.020 | 0.033 | 0.018 | 0.037 | 0.015 | 0.037 | 0.012 |
| $NF_{NH}$ | 69 | 17 | 53 | 16 | 47 | 13 | 41 | 11 |
| $HGF_{LH}$ | 1.31 | 0.02 | 1.35 | 0.03 | 1.37 | 0.03 | 1.38 | 0.04 |
| $\kappa_{LH}$ | 0.190 | 0.015 | 0.197 | 0.019 | 0.20 | 0.02 | 0.20 | 0.03 |
| $NF_{LH}$ | 31 | 17 | 47 | 16 | 53 | 13 | 59 | 11 |
| $VGF_{LV}$ | 0.96 | 0.02 | 0.96 | 0.02 | 0.97 | 0.02 | 0.96 | 0.02 |
| $NF_{LV}$ | 34 | 17 | 30 | 16 | 24 | 14 | 21 | 12 |
| $VGF_V$ | 0.49 | 0.05 | 0.52 | 0.04 | 0.54 | 0.04 | 0.54 | 0.04 |
| $NF_V$ | 66 | 17 | 70 | 16 | 76 | 14 | 79 | 12 |

The HGFs observed in Budapest correspond to the results reported for other urban areas (Cocker et al., 2001; Baltensperger et al., 2002; Ferron et al., 2005; Massling et al., 2005; Swietlicki et al., 2008 and references therein; Laborde et al., 2013; Lance et al., 2013; Ye et al., 2013, Cai et al., 2017); our data are slightly above the typical $HGF_{NH}$ and slightly below the typical $HGF_{LH}$ values (for a summary list see Swietlicki et al., 2008, Table 3). Laborde et al. (2013) further observed that the fresh traffic emissions have virtually hydrophobic behaviour, which is also close to our conclusion on the NH mode. The studies also reported the presence of a MH mode, which was not present in our data set. This mode is attributed in most cases either to the aged continental mostly soot-free background aerosol particles (Liu et al., 2013) entering the urban air or to particles from efficient biomass burning (BB; Swietlicki et al., 2008). The missing MH mode in the Budapest suggests that particles from local urban and rural regional sources seem more important inside the city than from continental background sources. This is in line with the decreasing tendency in the annual mean UF particle number concentration and SD from the city centre of Budapest ($8.4\pm5.3\times10^3$ cm$^{-3}$) through its near-city background ($3.1\pm2.8\times10^3$ cm$^{-3}$) to the more distant rural background ($3.8\pm3.6\times10^3$ cm$^{-3}$; Salma et al., 2014, 2016b), and with the estimated total concentrations for the continental background up to 500–800 cm$^{-3}$ (Raes et al., 2000). The distance of Budapest from the sea can possibly play a role. As far as the BB is concerned, it was recently estimated that the mean contribution and SD of OC from BB to the total carbon in central Budapest in winter was the largest single value of 34±8%, with contributions from fossil fuel combustion and biogenic emissions of 25±6% and 24±9%, respectively (Salma et al., 2017). The particles emitted from high-temperature efficient BB are rich in alkali salts and contain depleted amounts of organic compounds, thus they are expected to be very hygroscopic (Mircea et al., 2005; Rissler et al., 2005). The absence of the MH mode in Budapest could indirectly imply that the BB in the area likely took place as low-temperature incomplete combustion of biofuels, which produces organic aerosol constituents with limited water uptake. This hypothesis, however, needs to be further studied.

**4.3 Diurnal variations**

Mean diurnal variation of the number fractions of the NH and LV modes (Fig. 3) showed relatively large change during the day, and displayed a shape which corresponds to the typical daily activity-time pattern of inhabitants in cities, including particularly the road traffic in Budapest (Salma et al., 2011a). It consists of two peaks at approximately 08:00 and 18:00, which coincide with the most intensive vehicle traffic (morning and afternoon rush hours). This diurnal pattern of the modes was also strongly correlated with $N$ (correlation coefficients of $R$=0.907 and 0.882 for the NH and LV modes, respectively), and it is known that the total particle number concentration is mainly influenced by traffic emissions (Salma et al., 2014). It was also

found that the $NF_{LV}$ began to elevate already before the sunrise (in winter), and hence, before boundary layer mixing or photochemistry could intensively take place. This all implies that the NH and LV particles are related to the direct emissions from road vehicles.

Diurnal variation in the hygroscopicity was more complex. The mean diurnal dependency of the $\kappa$ value for all data, and separately for the NH and LH modes ($\kappa_{NH}$ and $\kappa_{LH}$, respectively) are shown in Fig. 4. The diurnal pattern of the $\kappa$ value for all data (Fig. 4a) showed the largest values during the early morning hours. This is different from those found in other studies in urban air. Lance et al. (2013) and Bialek et al. (2014) observed larger mean $\kappa$ value than in the present work, and its diurnal variation was different. The lowest hygroscopicity occurred in the early morning hours (around 05:00), followed by a steady increase until the early afternoon, when it started to decrease towards the early morning minimum. These differences can be likely explained by the fact that the locations of the previous studies cannot be regarded to be strictly city centre, and a strong natural/regional aerosol component could be present. This is further supported by the fact that the diurnal cycles they found are similar to those obtained for a remote site at Hyytiälä, Finland (Ehn et al., 2007). The diurnal hygroscopicity curves for the NH mode (Fig. 4b) were obviously anti-correlated with the traffic intensity, and with the man diurnal variation of $NF_{NH}$ (cf. Fig. 3a). This implies that the variation of $\kappa_{NH}$ was largely due to the diurnal variation of the $NF_{NH}$, thus the change was dominated by the emission sources. Our $\kappa$ values also agree well with the findings for fresh and aged particles emitted from a diesel engine (Tritscher et al., 2011). $\kappa_{LH}$ was larger during the daylight period than at night. This could be associated with the development of the boundary layer at noon, which can lead to dilution and can bring aged aerosol from the upper air parcels to the mixing layer (Cai et al., 2017). Similarly, photochemical reactions are generally more intensive around noontime. They promote aging processes, which lead to more oxidised organics and finally, to increased hygroscopicity. Its diurnal variation resembles the shape found in some other urban studies (Kitamori et al., 2009; Massling et al., 2005, Cai et al., 2017). For 50-nm particles, the LH mode displayed somewhat different shape, which indicated that these particles could have different composition from the other particles. Particles in this size range originate in large abundances from high-temperature primary emissions. It is also seen that the overall diurnal pattern of hygroscopicity (Fig. 4a) varied more with size than the individual NH (Fig. 4b) and LH modes (Fig. 4c). This implies that the changes in the $\kappa_{mean}$ are better explained by a changing number fraction of particles in each mode then by the size-dependent changes in chemical composition since the variation of $\kappa$ with size in the individual modes was smaller.

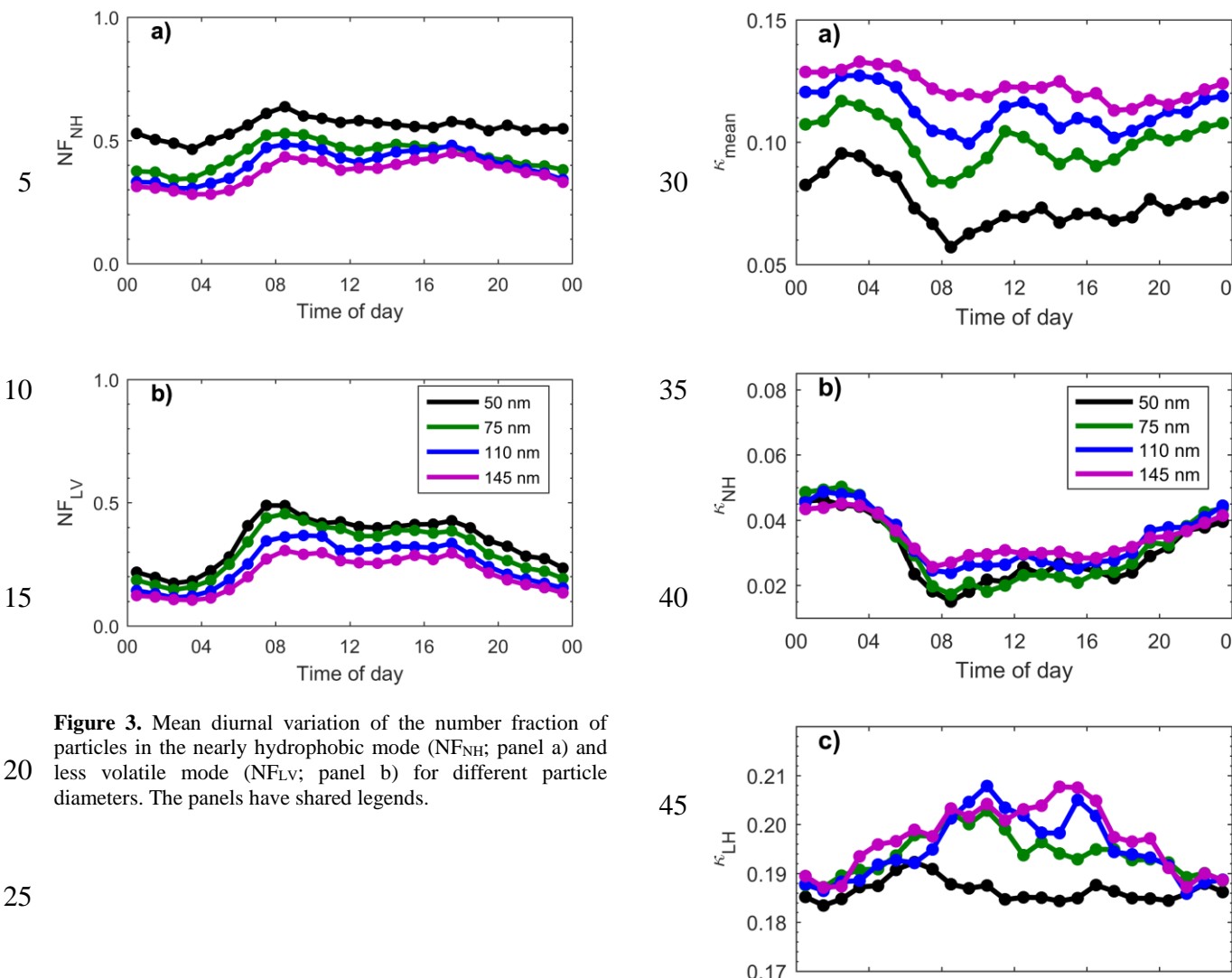

**Figure 3.** Mean diurnal variation of the number fraction of particles in the nearly hydrophobic mode (NF$_{NH}$; panel a) and less volatile mode (NF$_{LV}$; panel b) for different particle diameters. The panels have shared legends.

**Figure 4.** Mean diurnal variation of the hygroscopicity parameter κ for all data (κ$_{mean}$; panel a), separately for the nearly hydrophobic mode (κ$_{NH}$; panel b) and less hygroscopic mode (κ$_{LH}$; panel c) for different particle diameters. The panels have shared legends.

The VGF$_{LV}$ showed a clear daily activity-time pattern (Fig. 5a). This implies that the particles in the LV mode were mainly generated by vehicle road traffic. Particles with diameters of 50 and 70 nm seemed somewhat more volatile during the daylight time period than the 110- and 145-nm particles. The larger particles also had a larger magnitude in the VGF daily variation than the smaller particles. This indicates that the larger particles in the LV mode had more variation in their composition, and could consist of fresh non-volatile traffic emissions which collected or adsorbed condensing organics on their surface. For the VGF$_V$ (Fig. 5b), no obvious diurnal pattern was observed for any particle diameter. The associated particles showed more or less constant VGFs during the day. The smallest particles investigated (with a diameter of 50 nm) were separated from the others, and exhibited larger volatility (VGF<0.5) than for the other diameters, while particles with diameters of 110 and 145 nm showed almost identical volatile properties.

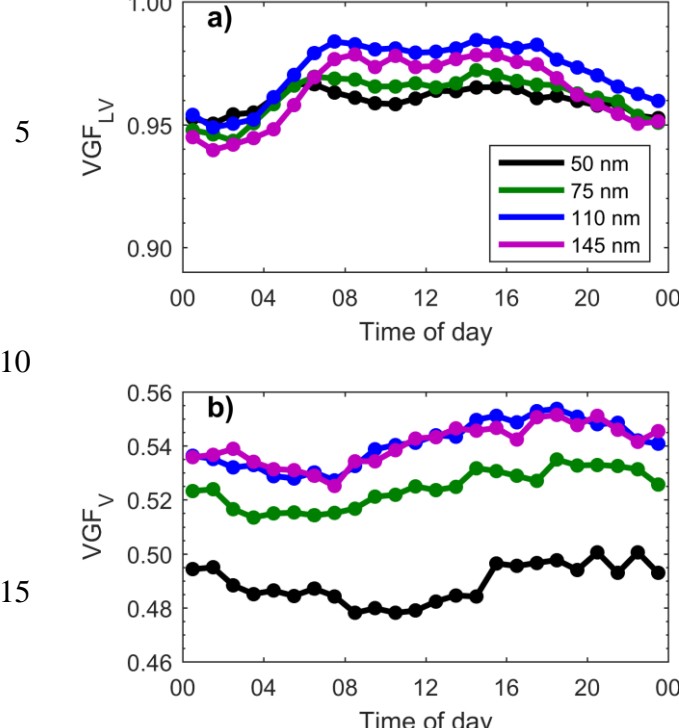

**Figure 5.** Mean diurnal variation of the volatility diameter growth factors for the less volatile mode (VGF$_{LV}$; panel a) and volatile mode (VGF$_V$; panel b) for different particle diameters. The panels have shared legends.

## 4.4 Relationships between workdays and weekends

The overall influence of vehicle emissions on the hygroscopicity and volatility of particles were studied by evaluating the data set separately for workdays and weekends. There is less vehicular road traffic during the weekends than on workdays (Salma et al., 2011a). Due to the limited residence time of particles with the selected diameters, their concentration levels on workdays and holidays also differ substantially because of the different source intensities. This facilitates the comparison of workdays and holidays. Diurnal variation of the total particle number concentration, fraction of particles in the volatile mode, and of the hygroscopicity parameter of the NH and LH modes for workdays and weekends are shown in Fig. 6. It is seen in Fig. 6a that the $N$ increased monotonically and rapidly from 05:30 to 07:00 on workdays, and reached its first maximum between 07:00 and 08:00. The concentration remained at elevated level over the whole daytime period. A second broader maximum appeared at around 18:00, and the $N$ decreased monotonically after this peak till about 04:00 next morning. On weekends, the morning growth was slower, and the first maximum was shifted to approximately 11:00, and its amplitude was much smaller. The remaining part of the curve was similar in shape to that for workdays but reached considerably smaller levels than on workdays. These are in good agreement with ordinary vehicular traffic flow in central Budapest (Salma et al., 2011a) except for the fact that the mean traffic flow from 00:00 to 05:00 is usually larger on weekends than on workdays. This latter difference can be likely explained by the changes in the composition of the vehicle fleet on workdays and weekends (less buses and heavy-duty vehicles on weekends and particularly overnights). Influence of the road traffic was clearly manifested in the diurnal variation of the number fraction of particles in the volatile mode (Fig. 6b), which essentially varied inversely with $N$ and traffic intensity. This probably indicated less variability in or constant shape of the $N_V$ concentration during the day. The extent of the decrease for the morning and evening rush hours decreased monotonically with the particle diameter. The hygroscopicity parameters for the LH mode on workdays or on weekends (Fig. 6c) were similar to each other. Except for the particles with a diameter of 50 nm, for which the $\kappa_{LH}$ was the smallest, they were basically comparable to each other in extent, and were without evident diurnal variation for both workdays and weekends. For the other diameters, the $\kappa_{LH}$ values were somewhat larger, and they

also showed larger values during the daylight time than during the night for both workdays and weekends, though their shape was mostly featureless. For weekends, the hygroscopicity parameters reached their largest value in the early morning hours, and seemed to be somewhat larger than for the workdays. The diurnal variation curves for the hygroscopicity of the LH mode were similar to each other separately for workdays and for weekends (Fig. 6d). The curves showed an inverse shape with $N$ and traffic intensity (cf. Fig. 6a). There was a larger decrease in the hygroscopicity parameter in the morning rush hours for workdays than for weekends, in particular for particles with diameters of 50 and 75 nm.

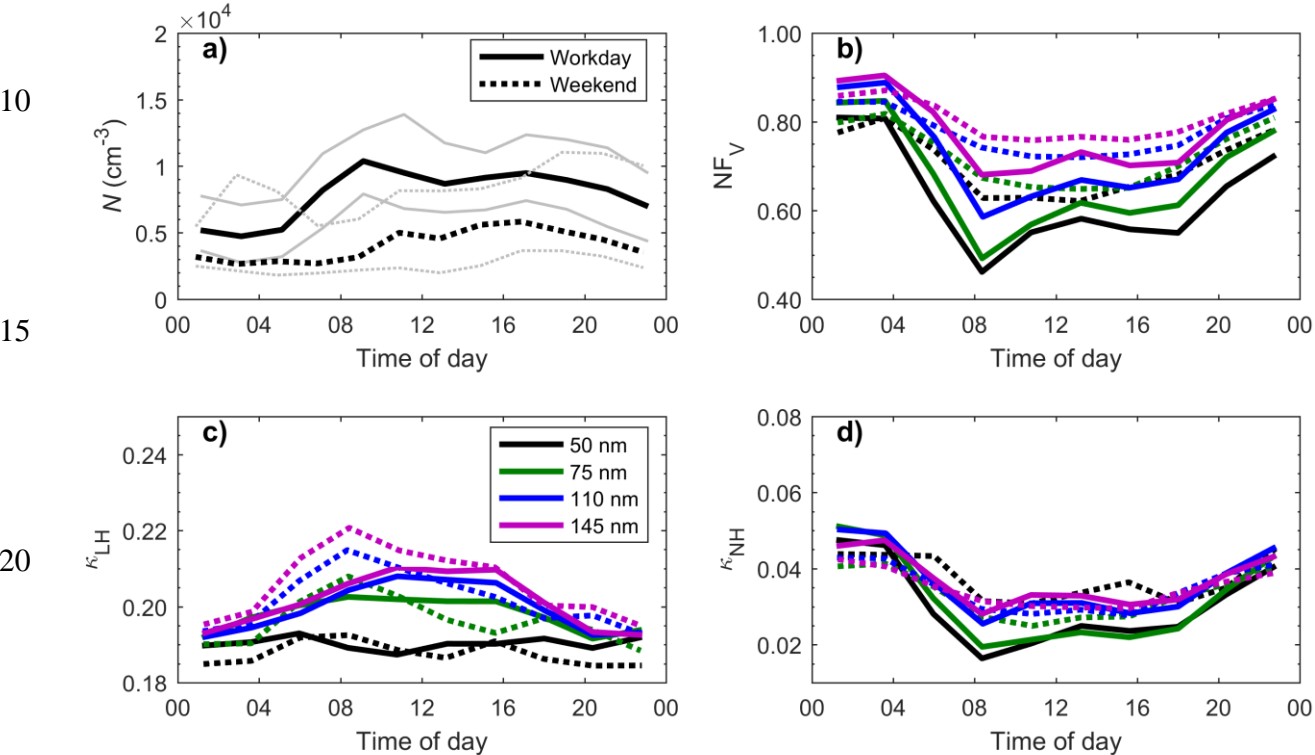

**Figure 6.** Diurnal variation of the median total particle number concentration ($N$; panel a), mean particle number fraction of volatile particles (NF$_V$; panel b), and the mean hygroscopicity parameter of the less hygroscopic mode ($\kappa_{LH}$; panel c) and nearly hydrophobic mode ($\kappa_{NH}$; panel d) separately for workdays (solid lines) and weekends (dotted lines). The thin lines in grey on panel a represent the lower and upper quartiles of the corresponding concentration data. The panels b, c and d have shared legends.

## 4.5 Conjugate hygroscopic and volatile properties

Relationships between hygroscopic and volatile properties were studied by relating their mean values to each other directly (Fig. 7 panel a) and by deriving the mean volume fraction remaining (VFR) from the particles after the thermal treatment as function of the mean HGF (Fig. 7 panel b). Both the VGF and VFR decreased monotonically with the hygroscopicity of particles. The relationships were size dependent, which may indicate that there were different abundances and/or different chemical species mainly of organic compounds in the particles with different dry diameters. A levelling off tendency was observed for particles with diameters of 110 and 145 nm at larger HGF values, and they also started to behave in a similar manner to each other. The VFR dependency for these two particle diameters seemed to be also limited from the bottom at approximately VFR=25%. This estimate is in good agreement with the conclusions obtained in an earlier study (Cheung et al., 2016), and seems to indicate an urban property. This all jointly suggests that the volatile and hygroscopic properties varied in a coherent manner, that the hygroscopic compounds were usually volatile, and that the larger particles contained internally mixed non-volatile chemical species in a considerable volumetric ratio as a refractory residual, which could be core-like soot or organic polymers.

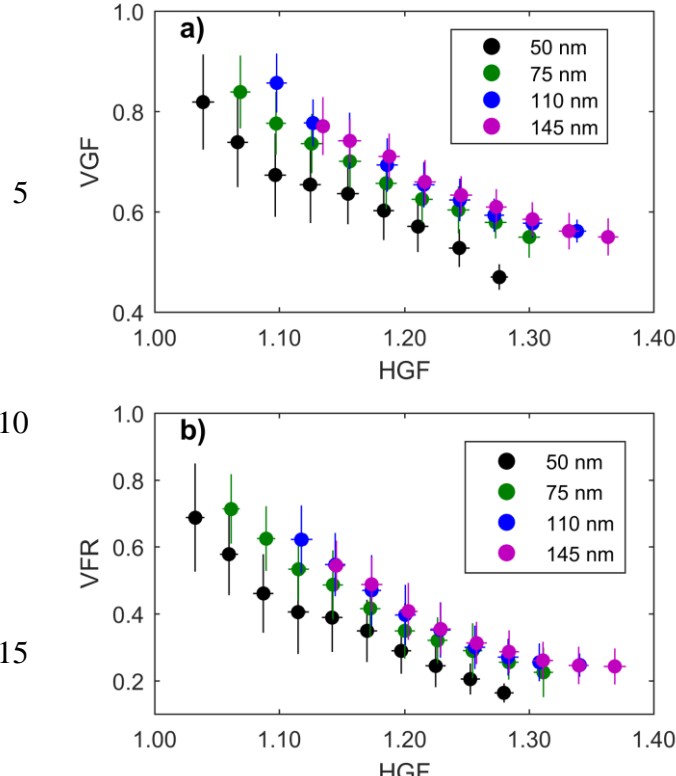

**Figure 7.** Relationship between the mean hygroscopic growth factors (HGFs) and the conjugate volatility growth factors (VGFs; panel a), and between the HGFs and the conjugate volume fraction remaining (VFR) after the thermal treatment (panel b) obtained at a mean relative humidity and at a mean temperature of 90% and 270 °C, respectively. The error bars indicate ±1 standard deviation.

There were further important links between the modes of the hygroscopicity and volatility probability density functions. Figure 8 shows the mean diurnal variation of κ-PDF and VGF-PDF for various dry diameters. It is worth mentioning that the colour coding depicts the normalized concentration fractions, and not the absolute values. For instance, the decrease in the volatile mode during the daytime was caused by appearance of less volatile particles in large numbers. The NH mode consisted of a mixture of particles with a κ<0.05 and a κ≈0.05. During the daytime interval, large numbers of small, very low hygroscopicity particles appeared in the NH mode, and became dominating the particle number concentrations for particles with diameters of 50 and 75 nm. Their influence was lower but still apparent for larger particles as well. Road traffic is the most likely source for these particles based on their size range and the timing of their appearance. Furthermore, the corresponding occurrence of the LV mode (Fig. 8 lower panels) supports the conclusion that road traffic was a major source of these particles. The NH mode for the larger particles was likely associated with a mixture of aged particles from traffic sources and biomass burning emissions as they are present during the day (Salma et al., 2017), and still show considerably low hygroscopicity. For the LH mode, the intermodal variation was less apparent. This mode showed a diurnal variation with a daytime maximum, which is similar to that observed for the overall hygroscopicity at rural sites (Ehn et al., 2007). The intermodal variation for VGFs can be explained by a constant background of volatile particles and by a varying contribution from less volatile particles from traffic emission. The observed decrease in VGF-PDF during daytime was caused by an increase in the total number of non-volatile particles, likely originating from traffic. This diminished the number fraction of the volatile mode, and hence, caused the decrease in the PDF, while the actual number concentration of particles in the volatile mode remained relatively stable.

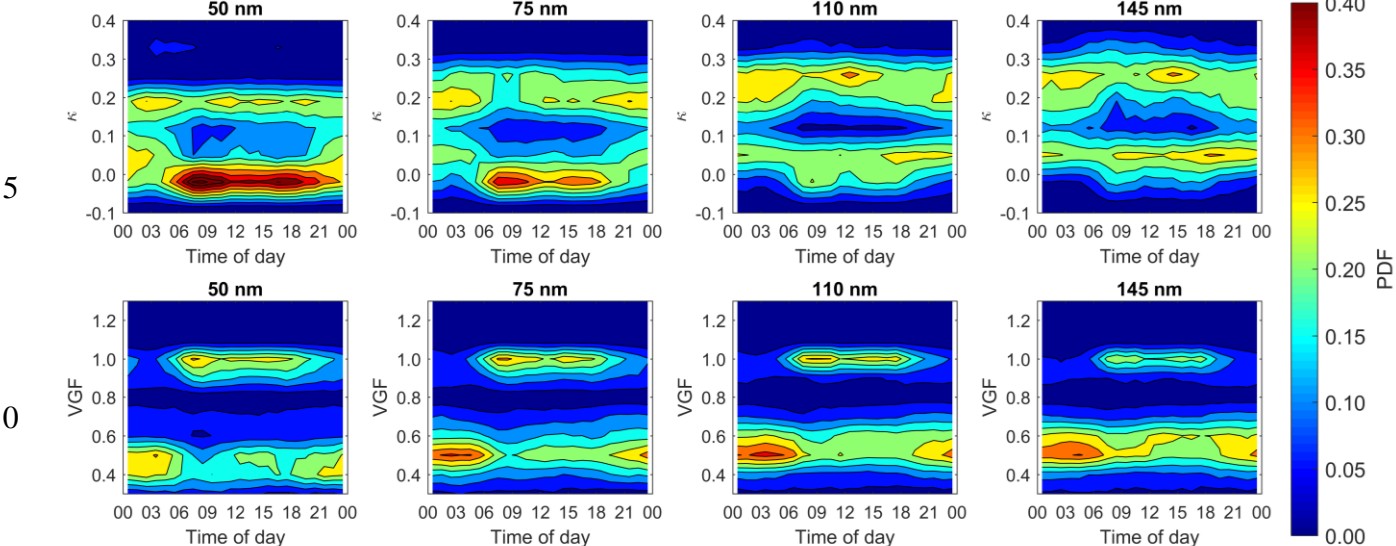

Figure 8. Mean diurnal variation of the hygroscopicity parameter κ and volatility growth factor (VGF) probability distribution functions (PDFs) for particles with dry diameters of 50, 75, 110 and 145 nm. The negative κ values are an artefact of plotting the data.

## 5 Conclusions

Hygroscopic GFs, volatility GFs and hygroscopicity parameters were quantified for ambient aerosol particles with dry diameters of (20,) 50, 75, 110 and 145 nm in situ by using a VH-TDMA system in central Budapest during two months in winter. The measurements were supported by a DMPS system and meteorological sensors, which were operated in parallel. The urban aerosol showed distinct bimodality with respect to both hygroscopic and volatile properties, which indicated that the urban aerosol contains an external mixture of particles with a diverse chemical composition. Vehicular road traffic had significant influence on both the hygroscopic and volatile properties, and contributed substantially to the particles in the NH and LV modes. These two modes were associated with each other, and both followed the typical diurnal pattern of road traffic and its workday/weekend variation. The LH mode was most likely composed of moderately transformed aged combustion particles consisting of partly oxygenated organics, inorganic salts and soot, and typically exhibited a VGF of approximately 0.6. Both the HGFs and VGFs showed modest size dependent behaviour, while the particles number fraction in the modes exhibited much stronger size dependency. Smaller particle diameters were associated with a larger number fraction of non-volatile and hydrophobic particles than the larger diameters. This can be explained by assuming that the larger particles grew by condensation of organic vapours, and it is also supported by the week dependency of the κ values with respect to the dry particle size. The 50-nm particles, however, had a considerably lower κ value and showed larger volatility during daytime in the LH mode with respect to the larger diameters. This suggests that these particles have different chemical composition than the larger particles. In general, the particles were mainly affected by local/urban emissions, and the aged particles were well separated from the freshly emitted ones.

The transformation process of soot particles from hydrophobic to hydrophilic in the real atmosphere is still not sufficiently understood and constrained. The present study emphasizes the importance of the mixing state of particles for influencing their hygroscopic properties. The ambient conditions during the campaign were typical for wintertime Budapest. Since there are strong seasonal variations in both the anthropogenic/natural and primary/secondary components, the hygroscopic and volatile properties are also expected to change over the year. Therefore, further similar in situ measurements should be carried out in different seasons in the future together with on-line chemical characterisation of particles to better quantify and understand the properties, relevance and role of urban aerosol.

The results and conclusion achieved in the present study cover a geographical region in Central Europe, which is much less represented and documented by similar measurements. This can already indicate a general value of our contribution. In addition to that, our findings could in principal be compared to earlier studies performed specifically in urban environments in the word. Nevertheless, there are substantial differences among cities and their associated larger regions, which also means that a perspective comparative analysis should be detailed enough, it is expected to be extended and more importantly, to deal with systematic methods of identifying and quantifying similarities and differences. The present work indicates the need for such an overview work in the future if the number of cities with available corresponding data further increases.

## 6 Data availability

The observational data used in this paper are available on request from J. E.

*Competing interests.* The authors declare that they have no conflict of interest.

*Acknowledgements.* Financial support by the National Research, Development and Innovation Office, Hungary (contracts K116788 and PD124283), by the European Regional Development Fund and the Hungarian Government (GINOP-2.3.2-15-2016-00028), by the European Commission H2020 research and innovation program via ACTRIS-2 (grant agreement 654109), by the European Research Council (ERC-advanced grant, ATM-GTP) and by the Academy of Finland, Center of Excellence (project 307331) and Academy Professor Project (M. K.) are gratefully acknowledged.

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
