# Peer review of "Wintertime hygroscopicity and volatility of ambient urban aerosol particles"

_Atmospheric Chemistry and Physics, 2017_

## Referee Comment (RC1) · Anonymous Referee #3 · 12 Dec 2017

Enroth et al. reported the hygroscopic and volatile properties of atmospheric aerosol particles with varying dry diameters by using a VH-TDMA system in central Budapest during two months in winter. The urban particles showed distinct bimodality with respect to both hygroscopic and volatile properties, which were significantly influenced by vehicular road traffic. While this paper uses sound techniques and is generally well written, substantial revisions especially for the discussion section are needed before this manuscript can be considered for publication in ACP. Major comment: The authors did not discuss their data and results adequately within the framework of current knowledge in the literature. Thus, it is not clear to the readers how atmospherically important of this work is and what new information it has offered. The authors could expand the discussion section by framing the results in this study into the existing

literature to highlight the contribution to scientific progress. In addition to this very general comment, several specific comments and some requested clarifications are outlined below. Specific comment: P1 L7: "atmospheric aerosol particles" should be "atmospheric particles" or "aerosol particles". P1 L12: "it was decreasing monotonically from 71% to 41% with particle diameter." With INCREASING OR DECREASING particle diameter? It should be expressed accurately. The corresponding modifications should be made throughout the whole manuscript. For example, P1 L17 and P1 L19. P1 L28-31: The last sentence in the abstract is rather tedious. It should be rephrased to make it clear. I suggest the authors check throughout the manuscript as there are a few other cumbersome statements. P2 L9-12: The sentence should be rephrased or broken into two. P2 L26-27: As is listed, there are many references on the measurements on complex urban aerosols. Why the authors stated that corresponding measurements are so scarce? P2 L29: This sentence should be rephrased to make it clear. P3 L17: "using a silica-gel diffusion dried at indoor temperatures". "dried" should be deleted. P3 L29: This temperature was selected by considering previous experience. The authors should clarify how the temperature was selected based on previous experience. P5 L3: It is not clear which surface tension value ($\sigma$ = 72 mN m$^{-1}$ or 60 mN m$^{-1}$) was used in the calculations. Please clarify it in the manuscript. P5 L6: Please clarify the experimental uncertainties in detail. P5 L28: What is the size range for UF particles? P6 L1: The contribution of the two modes was size dependent. It is not clear how it can be concluded from Fig.1 as only data for particles with a dry diameter of 145 nm was given.

Please also note the supplement to this comment:
https://www.atmos-chem-phys-discuss.net/acp-2017-926/acp-2017-926-RC1-supplement.pdf
* * *

---

## Referee Comment (RC2) · Anonymous Referee #2 · 16 Dec 2017

This manuscript by Enroth et al. entitled as 'Wintertime hygroscopicity and volatility of ambient urban aerosol particles' reports size-segregated hygroscopicity and volatility measurements of ambient particles at Budapest. The technical quality of the measurement looks reasonably good (at least, comparable to former studies conducted at different places in the world). This manuscript could add one data point to a map for global distributions on physical properties of aerosol particles. I have the following comments.

Major comments: Although authors showed the data, it was not clear to me how the data of the present study could be compared with former studies at other urban areas. If I understand it correctly, the uniqueness and novelty of the study is that the authors have conducted an atmospheric observation at Budapest in winter. So, it would be

important to compare if the data at Budapest is similar to/different from those in other cities.

Specific comments: Title: the title of the manuscript could be modified so that it contains more detailed information (e.g., adding information about the observation site).

Abstract: The current abstract is too long (more than 450 words). If I remember correctly, this journal does not have any length limit for abstract. However, I believe that the abstract could be shortened by almost 50% if it is described concisely.

HTDMA measurements: The authors state that the RH stability of the measurement was $90 \pm 2\%$ (standard deviation). This value is not so small, compared with other HTDMA systems (e.g., [Duplissy et al., 2009]). It would be ideal to have a description on how the fluctuation in RH could influence interpretation of observation data.

Application of TDMAinv on VTDMA data: I understand that the method was developed mainly for analysis of HTDMA data. Unlike HTDMA, a part of particles passing through DMA1 completely evaporate (i.e., disappear) by heat during thermal desorption process[Kuwata and Kondo, 2008]. I wonder how this type of particles was considered during the data inversion process.

VFR: Although VFR is frequently used for volatility study on bulk aerosol particles, I am not sure if it is commonly used for VTDMA study. Would it be possible to explain why this metric is useful in obtaining a physically meaningful parameter?

P8L4: 'while their coating with succinic acid, sulphuric acid or polyaromatic hydrocarbons (PAHs) influenced the hygroscopic growth in a complex way.' PAHs are not hygroscopic at all. Would it be possible to clarify how they could influence hygroscopic growth?

P9L23: 'Since the rush hours also coincided with the sunrise and sunset in winter' Is there any supporting evidence for this statement?

References Duplissy, J., et al. (2009), Intercomparison study of six HTDMAs: results

and recommendations, Atmos. Meas. Tech., 2(2), 363-378, doi:10.5194/amt-2-363-2009. Kuwata, M., and Y. Kondo (2008), Dependence of size-resolved CCN spectra on the mixing state of nonvolatile cores observed in Tokyo, J. Geophys. Res., 113(D19), doi:10.1029/2007jd009761.

---

## Referee Comment (RC3) · Anonymous Referee #1 · 17 Dec 2017

This manuscript by Enroth et al. presents some valuable measurement results regarding an important issue "hygroscopic and volatile properties of urban aerosol particles. This study reported the size-resolved hygroscopic and volatile growth factor of particles at Budapest, and discussed its diurnal variation and the difference between workday and weekend. This paper aims to provide important parameters (hygroscopic and volatile growth, mixing state) for understanding the atmospheric aging processes and anthropogenic activity. However, the paper does not bring much real novel findings (or understanding) to the urban aerosol hygroscopicity and volatility. While this paper is generally well written, substantial revisions are needed before it can be considered for publication in ACP, especially for the discussion section. Major comments: 1. My major concern with this work is that some speculative conclusions are drawn from the

measurements that are currently not supported (some instances are listed in the specific comments). The author should provide further evidences (or data) to support their conclusions or more extensive discussion, otherwise they should be removed. 2. The presentation of data should be improved. There were serval parameters used in the article to describe particles hygroscopicity and volatility. In order to avoid any confusion, I suggest that the authors use HGF and VGF to present particles hygroscopic and volatile growth factor in the body text and figures. The subscript like $\kappa$LH is also suggested. 3. It would be advisable to discuss the impact of anthropogenic activities on aerosol hygroscopicity and volatility in detail. The comparison between workday and weekend is quite interesting. However, it still lacks some deep investigation. For example, it would be very interesting to know how the relationship between the number fraction and growth factor of NH and LV would change in workday and weekend. The comparison of the diurnal variation of particle number size distribution was also an interesting way in this section. 4. Although authors showed the data at Budapest, it was not very clear to me how the results of this study could be compared with previous studies at other urban areas. Specific comments: 1 P4 Line 12: Please add a citation about the earlier study here. 2 P4 Line 34: Please add the unity of each parameter. 3 P4 Line 36: It would be better to use Kelvin temperature instead of Celsius temperature here. 4 P4 Line 39: Is the depression of surface tension only controller by HULIS? 5 P5 Line 1-4: Is there other organics matter can reach the thermodynamic equilibrium fast enough to depress the surface tension in the humidifier tube. 6 P5 Line 8: Please explain why the influence of surface tension is smaller under subsaturated condition. 7 P5 Line 17: Why the parameter "volume fraction remaining" was used here to describe the particle volatility? The parameter "volatile growth factor" is quite enough in discussing particle volatility. 8 P5 Line 16: Please specify the actual value of RH when measuring the GF. 9 P5 Line 28: What is the meaning of "N" here? Do you mean total particles? It was a little bit confused. Please change the expression. 10 P5 Line 37: It may be not necessary to mention the reason of missing data. 11 P6 Line 3-4: According to Fig. 1, these two modes did not seem such distinct on January 16th, and

the particle number concentration decreased on the same time. It would be interesting to analyze the data during this period. 12 P6 Line 30: According to former section, the RH in the humidifier was set to 85% at the beginning of measurement and then set to 90%. Please specific the GF-PDF was measured under which RH value. Please use HGF-PDF and VGF-PDF in this article. 13 P6 Line 34: In sentence "... the median ambient concentration of these particles ..." Please add the actual value of the number concentration in this sentences. 14 P7 Line 30: The author should note the equation used in calculating mean growth factor and number fraction. 15 P7 Line 31: In order to avoid any confusion, please use "number fraction" instead of "relative concentration" in the discussion. 16 P7 Line 37: "The slight change in the hygroscopic growth behaviour cannot be reliably interpreted without measurements at high (>95%) RHs". Please give a more specific discussion on this sentence. Why the higher RH measurement was needed here? 17 P8 Line 17-19: I don't think that the GF for pure NH4NO3 and (NH4)2SO4 higher than the measured value indicates the LH species become coated with inorganics species. It is not strong evidence. Please modify this sentence. 18 P8 Line 21: Please use the full name instead of "TED/EDS". 19 P8 Line 34: "Our data are slightly below the average GFs". Please list the value of average GFs. 20 P9 Line 7: Please use standard deviation instead of "SD". 21 P9 Line 21-23: The correlation co-efficients less than zero suggests that the number fraction of NH and LV modes were negative correlated with the total particle number concentration. Please explain the relationship between these negative correlations with the traffic emissions. 22 Figure 4: Please use $\kappa$mean, $\kappa$NH and $\kappa$LH in the label of y-axis. 23 P9 Line 41-42: The $\kappa$LH was larger during daytime could result from the development of boundary layer at noon, which would be able to bring aged aerosol from upper atmosphere layer to the ground. 24 P10 L1-2: Because 50 nm particles were more originated from primary emission. It would be better to be mentioned in the discussion. 25 P10 L4-5: The changes in chemical composition including mass distribution for a certain particle size would lead to the change of number fraction in each mode and mean $\kappa$. The sentence should be modified. 26 Figure 5: Please use VGF in the label of y-axis. 27 P11 Line

36-37:". . . and since the atmospheric residence time of the particles with the selected diameters is estimated to be several hours." What is relationship between the particles residence time and separating the data? This sentence was not necessary. Please delete it or try to explain more. 28 P12 Line10-11: ". . . but without substantial variation during the daylight period". It might be wrong. According to fig. 6 (c), the $\kappa$ values increased during daytime and became much higher than in nighttime. Please explain that. 29 Section 4.5: There is not essentially difference between VGF and VFR. Please delete the part of VFR and focus on the relationship between VGF and HGF. 30 Section 4.5: It would be interesting to discuss the relationship between VGF and HGF for each mode. 31 P13 Line 34-35: "The smallest particles (with dry diameters of 50 and 75 m) appeared to be dominated by vehicle emissions since their size range, diurnal variability and timing matched the traffic intensity and 35 emissions." It doesn't consist with the former discussion in page 11 line 2-3 "Particles with diameters of 50 and 70 nm seemed more volatile during the daylight time period than the 110- and 145-nm particles. The latter two sizes are in the typical size range of fresh diesel emissions (Charron and Harrison, 2003)". Please explain that.

References: Cai, M. et al, Comparison of Aerosol Hygroscopcity, Volatility, and Chemical Composition between a Suburban Site in the Pearl River Delta Region and a Marine Site in Okinawa. Aerosol and Air Quality Research. Doi:10.4209/aaqr.2017.01.0020

---

## Author Comment (AC1) · 14 Feb 2018

**Response to Referee #3**

The authors would like to thank Referee #3 for his/her detailed and valuable comments to further improve and clarify the MS. We have considered all recommendations, and made the appropriate alterations. The changes can be explicitly tracked in the annotated version of the MS. Our specific responses to the comments are as follows.

Major comment

The authors did not discuss their data and results adequately within the framework of current knowledge in the literature. Thus, it is not clear to the readers how atmospherically important of this work is and what new information it has offered. The authors could expand the discussion section by framing the results in this study into the existing literature to highlight the contribution to scientific progress. In addition to this very general comment, several specific comments and some requested clarifications are outlined below.

We revised thoroughly and improved the Results and discussion section at several places and from several aspects with more detailed arguments. We also modified the body text to make our intentions and statements more detailed, specific, and further literature sources were also included to support our conclusions.

Specific comments

P1 L7: "atmospheric aerosol particles" should be "atmospheric particles" or "aerosol particles".

The formulation "atmospheric aerosol particles" was selected to express that we investigated aerosol particles in the ambient air with a larger and open spatial scale, and that we did not confined our study to specific or more closed urban environments. This is an ordinary concept in aerosol science manifested in several key textbooks such as e.g. Hinds, C. W.: Aerosol Technology, Wiley, 1999, chapter 14 or Seinfeld, J. H. and Pandis, S. N.: Atmospheric Chemistry and Physics, Wiley, 1998, chapter 7.

P1 L12: "it was decreasing monotonically from 71% to 41% with particle diameter." With INCREASING OR DECREASING particle diameter? It should be expressed accurately. The corresponding modifications should be made throughout the whole manuscript. For example, P1 L17 and P1 L19.

An increase or decrease of a function with an independent variable expresses the change caused by an increasing tendency in the variable. This type of the possible formulations is indeed a simplification, but it avoids disturbing over-detailed wording and repetitions if one gets used to it. We chose and adopted this, progressively acknowledged convention consequently throughout the MS.

P1 L28-31: The last sentence in the abstract is rather tedious. It should be rephrased to make it clear. I suggest the authors check throughout the manuscript as there are a few other cumbersome statements.

The abstract was revised and restructured substantially including the last sentence. In addition, we checked thoroughly and modified some other sentences to clarify and improve their meaning.

P2 L9-12: The sentence should be rephrased or broken into two.

The sentence was split into 2 sentences.

P2 L26-27: As is listed, there are many references on the measurements on complex urban aerosols. Why the authors stated that corresponding measurements are so scarce?

The sentence was modified to express our intention more clearly, and new recent references were also added.

P2 L29: This sentence should be rephrased to make it clear.

The sentence was reformulated.

P3 L17: "using a silica-gel diffusion dried at indoor temperatures". "dried" should be deleted.

The word "dried" was misspelled, and it was corrected to "dryer" now.

P3 L29: This temperature was selected by considering previous experience. The authors should clarify how the temperature was selected based on previous experience.

We included more details on this argument.

P5 L3: It is not clear which surface tension value ($\sigma$=72 mN m$^{-1}$ or 60 mN m$^{-1}$) was used in the calculations. Please clarify it in the manuscript.

The calculations were performed by using the surface tension of pure water, thus 72 mN m$^{-1}$. A brief sentence was added to clarify this.

P5 L6: Please clarify the experimental uncertainties in detail.

The estimated uncertainty was added.

P5 L28: What is the size range for UF particles?

The size range of UF particles ($d$<100 nm) was specified at its first occurrence in the body text (on page 2).

P6 L1: The contribution of the two modes was size dependent. It is not clear how it can be concluded from Fig.1 as only data for particles with a dry diameter of 145 nm was given.

The size dependence does not follow from the Fig. 1. It can be inferred from Table 1. We modified the related sentence, and added some new information on this for the clarification.

Imre Salma
14–02–2018

---

## Author Comment (AC2) · 14 Feb 2018

**Response to Referee #2**

The authors would like to thank Referee #2 for his/her detailed and valuable comments to further improve and clarify the MS. We have considered all recommendations, and made the appropriate alterations. The changes can be explicitly tracked in the annotated version of the MS. Our specific responses to the comments are as follows.

Major comment

Although authors showed the data, it was not clear to me how the data of the present study could be compared with former studies at other urban areas. If I understand it correctly, the uniqueness and novelty of the study is that the authors have conducted an atmospheric observation at Budapest in winter. So, it would be important to compare if the data at Budapest is similar to/different from those in other cities.

We revised thoroughly and improved the Results and discussion section at several places and from several aspects with more detailed arguments. We also modified the body text to make our intentions and statements more detailed, specific, and further literature sources were also included to support our conclusions.

Specific comments

Title: the title of the manuscript could be modified so that it contains more detailed information (e.g., adding information about the observation site).

The hygroscopic and volatile properties of particles in winter and the results/conclusions obtained from them contribute to the improved understanding of the urban-type atmospheric environment and not just of a specific city. The present title emphasises the value of the VH-TDMA measurements in cities in general. This was our motivation for the present formulation of the title. The details of the study including the observation site are given exactly already in the abstract, and further in body text.

Abstract: The current abstract is too long (more than 450 words). If I remember correctly, this journal does not have any length limit for abstract. However, I believe that the abstract could be shortened by almost 50% if it is described concisely.

The abstract was revised and restructured substantially, and it was also shorten as requested.

HTDMA measurements: The authors state that the RH stability of the measurement was 90 ± 2% (standard deviation). This value is not so small, compared with other HTDMA systems (e.g., [Duplissy et al., 2009]). It would be ideal to have a description on how the fluctuation in RH could influence interpretation of observation data.

As a result of this comment, we realised by double checking that the standard deviation of the mean RH and mean denuder $T$ were given in an incorrect way as a result of faulty data handling (as the maximal deviations). The correct values were given. They comply with the ordinary uncertainty range described in Duplissy et al., 2009.

Application of TDMAinv on VTDMA data: I understand that the method was developed mainly for analysis of HTDMA data. Unlike HTDMA, a part of particles passing through DMA1 completely evaporate (i.e., disappear) by heat during thermal desorption process [Kuwata and Kondo, 2008]. I wonder how this type of particles was considered during the data inversion process.

The volatility properties of particles with a diameter of 20 nm were not interpreted because they appeared at the lower end of the VGF range after the shrinkage, and thus their diameters were close to the detection limit of the CPC used as the detector, and they were also subjected to enhanced diffusional losses. The complete evaporation of particles with a diameter >50 nm was not considered in this study. A comment on this, and the estimation of the magnitude of the loss was briefly discussed.

VFR: Although VFR is frequently used for volatility study on bulk aerosol particles, I am not sure if it is commonly used for VTDMA study. Would it be possible to explain why this metric is useful in obtaining a physically meaningful parameter?

The two terms, namely the volatility growth factor and the volume fraction remaining have related meanings. The former quantity shows the diameter change, while the latter property represents a very expressive picture on the physical appearance of the coating and core of particles, and on their volume ratio. Therefore, we would like to keep this in the MS as well as an auxiliary property.

P8L4: 'while their coating with succinic acid, sulphuric acid or polyaromatic hydrocarbons (PAHs) influenced the hygroscopic growth in a complex way.' PAHs are not hygroscopic at all. Would it be possible to clarify how they could influence hygroscopic growth?

According to the cited reference, "PAHs are slightly soluble or even insoluble in water and the solubility decreases with increasing molecular mass. The comparison of the mass spectra of untreated soot and sulfuric acid coated soot showed that the fraction of mass peaks with m/z>150 dropped and the fraction with m/z<150 increased. Probably the sulfuric acid reacts

with the PAHs and forms products with lower molecular mass than the initial PAHs. (…) Reactions of sulfuric acid with PAH have been previously observed. (…) These products could have a higher solubility in water than the initial PAHs. As a consequence, the hygroscopic growth and activated fraction increased due to (a) the products of this reaction and (b) the unconsumed coating fraction itself." We appreciate the comment, and added this specific information as a further explanation and clarification.

P9L23: 'Since the rush hours also coincided with the sunrise and sunset in winter' Is there any supporting evidence for this statement?

The sentence was removed.

Imre Salma
14–02–2018

---

## Author Comment (AC3) · 14 Feb 2018

**Response to Referee #1**

The authors would like to thank Referee #1 for his/her detailed, extensive and valuable comments to further improve and clarify the MS. We have considered all recommendations, and made the appropriate alterations. The changes can be explicitly tracked in the annotated version of the MS. Our specific responses to the comments are as follows.

Major comments

1. My major concern with this work is that some speculative conclusions are drawn from the measurements that are currently not supported (some instances are listed in the specific comments). The author should provide further evidences (or data) to support their conclusions or more extensive discussion, otherwise they should be removed.

We revised thoroughly and improved the Results and discussion section at several places and from several aspects with more detailed arguments. We also modified the body text to make our intentions and statements more detailed, specific, and further literature sources were also included to support our conclusions.

2. The presentation of data should be improved. There were serval parameters used in the article to describe particles hygroscopicity and volatility. In order to avoid any confusion, I suggest that the authors use HGF and VGF to present particles hygroscopic and volatile growth factor in the body text and figures. The subscript like $\kappa\_LH$ is also suggested.

We adopted all these suggestions at several places throughout the abstract, body text and figures.

3. It would be advisable to discuss the impact of anthropogenic activities on aerosol hygroscopicity and volatility in detail. The comparison between workday and weekend is quite interesting. However, it still lacks some deep investigation. For example, it would be very interesting to know how the relationship between the number fraction and growth factor of NH and LV would change in workday and weekend. The comparison of the diurnal variation of particle number size distribution was also an interesting way in this section.

We completed new investigations in the requested sense, and their results were included in the discussions.

4. Although authors showed the data at Budapest, it was not very clear to me how the results of this study could be compared with previous studies at other urban areas.

We added new aspects and reformulated several sentences to place our results in an international framework in a more extensive manner.

Specific comments

P4 Line 12: Please add a citation about the earlier study here.

Several new reference was added.

P4 Line 34: Please add the unity of each parameter.

The actual values utilised in the calculations were explicitly given. The units of the parameters comply with SI units according to the Mathematical notation and terminology guidelines of the journal.

P4 Line 36: It would be better to use Kelvin temperature instead of Celsius temperature here.

The data was converted to Kelvin unit.

P4 Line 39: Is the depression of surface tension only controller by HULIS?

Atmospheric humic-like substances are reported to be the most abundant and important surface active aerosol component (e.g. Facchini et al., 1999; Fuzzi et al., 2001). They can decrease the surface tension of water substantially (Decesari et al., 2001; Salma et al., 2006), and this is ordinarily considered to be the most important contribution of the depression. The related sentence was reformulated to be more precise, and to include jointly the new aspects raised in Comments 4–6.

P5 Line 1-4: Is there other organics matter can reach the thermodynamic equilibrium fast enough to depress the surface tension in the humidifier tube.

A number of empirical relationships were reported to relate surface tension to concentration of OC or WSOC in droplets (e.g. Facchini et al., 1999). These were, however, suggested mainly for predicting the effect of organics on cloud activation. A full consideration of the effect in multicomponent aerosol systems has not been reported yet. It is expected, however, that such effects can be important in H-TDMA studies on nucleation mode particles ($<\approx$25 nm; Swietlicki et al., 2008). Therefore, the effects of organic compounds other than HULIS were not considered in the present study. The related sentence was reformulated to include jointly the new aspects raised in Comments 4–6.

P5 Line 8: Please explain why the influence of surface tension is smaller under subsaturated condition.

The sensitivity of hygroscopic growth to the surface tension becomes more important with decreasing dry particle diameter and increasing RH since these are the conditions under which HGF is most sensitive to the Kelvin factor. The change in the Kelvin term caused by the altered surface tension in sub-saturated conditions is smaller when compared to its dependency in cloud activation. The related sentence was reformulated to include jointly the new aspects raised in Comments 4–6.

P5 Line 17: Why the parameter "volume fraction remaining" was used here to describe the particle volatility? The parameter "volatile growth factor" is quite enough in discussing particle volatility.

The two terms, namely the volatility growth factor and the volume fraction remaining have related meanings. The former quantity shows the diameter change, while the latter property represents a very expressive picture on the physical appearance of the coating and core of particles, and on their volume ratio. Therefore, we would like to keep this in the MS as well as an auxiliary property.

P5 Line 16: Please specify the actual value of RH when measuring the GF.

The actual RH corresponding to the HGF was added.

P5 Line 28: What is the meaning of "N" here? Do you mean total particles? It was a little bit confused. Please change the expression.

The abbreviation $N$ was explained in page 5 line 23 of the original MS (and in the table/figure captions as well). We added further textual formulation now to increase its visibility.

P5 Line 37: It may be not necessary to mention the reason of missing data.

The part of the sentence was removed as requested.

P6 Line 3-4: According to Fig. 1, these two modes did not seem such distinct on January 16th, and the particle number concentration decreased on the same time. It would be interesting to analyze the data during this period.

The HGFs for the NH and LH modes were the closest to each other in the very beginning of 16 January during the time interval in which the atmospheric concentrations were low. Nevertheless, they could be well resolved. The effect was related to data fluctuations and possible effects of local meteorology.

P6 Line 30: According to former section, the RH in the humidifier was set to 85% at the beginning of measurement and then set to 90%. Please specific the GF-PDF was measured under which RH value. Please use HGF-PDF and VGF-PDF in this article.

The actual RH corresponding to the HGF was explicitly given in the text. The suggested notifications were adopted.

P6 Line 34: In sentence "... the median ambient concentration of these particles ..." Please add the actual value of the number concentration in this sentences.

The median concentration was expressed numerically as well as (in addition to Fig. 2a) as requested.

P7 Line 30: The author should note the equation used in calculating mean growth factor and number fraction.

Short descriptions with the equations were added (Eqs. 2 and 3.)

P7 Line 31: In order to avoid any confusion, please use "number fraction" instead of "relative concentration" in the discussion.

The requested change was adopted at several places in the body text and table/figure captions.

P7 Line 37: "The slight change in the hygroscopic growth behaviour cannot be reliably interpreted without measurements at high (>95%) RHs". Please give a more specific discussion on this sentence. Why the higher RH measurement was needed here?

The differences in the HGF are expected to become larger with RH in the sub-saturated interval, so the increased RH to 95% or even higher are expected to yield more evident or more reliable differences. The related sentence was reformulated.

P8 Line 17-19: I don't think that the GF for pure NH4NO3 and (NH4)2SO4 higher than the measured value indicates the LH species become coated with inorganics species. It is not strong evidence. Please modify this sentence.

The sentence was removed.

P8 Line 21: Please use the full name instead of "TED/EDS".

The abbreviation TEM/EDS was extended and resolved as transmission electron microscopy with electron energy-loss spectroscopy (TEM/EELS).

P8 Line 34: "Our data are slightly below the average GFs". Please list the value of average GFs.

The sentence was extended to clarify our intension, and a comprehensive reference was added.

P9 Line 7: Please use standard deviation instead of "SD".

The abbreviation SD was accepted or even promoted in our recent ACP papers in the copy-edition phase as it conforms the English guidelines and house standards of the journal.

P9 Line 21-23: The correlation coefficients less than zero suggests that the number fraction of NH and LV modes were negative correlated with the total particle number concentration. Please explain the relationship between these negative correlations with the traffic emissions.

The negative signs were simply a typing mistakes, and they were removed.

Figure 4: Please use $\kappa\_mean$, $\kappa\_NH$ and $\kappa\_LH$ in the label of y-axis.

The subscripts were added to the figures as requested.

P9 Line 41-42: The $\kappa\_LH$ was larger during daytime could result from the development of boundary layer at noon, which would be able to bring aged aerosol from upper atmosphere layer to the ground.

This effect was also added to the discussion as an important possibility, and a new reference was given.

P10 L1-2: Because 50 nm particles were more originated from primary emission. It would be better to be mentioned in the discussion.

We extended the text to include this information explicitly.

P10 L4-5: The changes in chemical composition including mass distribution for a certain particle size would lead to the change of number fraction in each mode and mean κ. The sentence should be modified.

The sentence was extended accordingly.

Figure 5: Please use VGF in the label of y-axis.

The axis label was modified as requested.

P11 Line 36-37: "… and since the atmospheric residence time of the particles with the selected diameters is estimated to be several hours." What is relationship between the particles residence time and separating the data? This sentence was not necessary. Please delete it or try to explain more.

Due to the limited residence time of particles with the investigated diameters, they remain in the air for several hours only, and therefore, their concentration levels on workdays and holidays usually differ substantially because of their different sources intensities during these time intervals. It is this basic property that makes the comparison of the workdays and holidays sensible. The statement was separated from the first part of the sentence, and largely reformulated to avoid the misunderstanding.

P12 Line10-11: "… but without substantial variation during the daylight period". It might be wrong. According to fig. 6 (c), the κ values increased during daytime and became much higher than in nighttime. Please explain that.

The sentence dealt with the daylight period only. Nevertheless, we reformulated and extended it to finish its misleading character.

Section 4.5: There is not essentially difference between VGF and VFR. Please delete the part of VFR and focus on the relationship between VGF and HGF.

The two terms, namely the volatility growth factor and the volume fraction remaining have related meanings. The former quantity shows the diameter change, while the latter property represents a very expressive picture on the physical appearance of the coating and core of particles, and on their volume ratio. Therefore, we would like to keep this in the MS as well as an auxiliary property. Nevertheless, we shortened its discussion as requested.

Section 4.5: It would be interesting to discuss the relationship between VGF and HGF for each mode.

New aspects of the comparison of VGF and HGF separately for the modes was added to the body text.

P13 Line 34-35: "The smallest particles (with dry diameters of 50 and 75 m) appeared to be dominated by vehicle emissions since their size range, diurnal variability and timing matched the traffic intensity and 35 emissions." It doesn't consist with the former discussion in page 11 line 2-3 "Particles with diameters of 50 and 70 nm seemed more volatile during the daylight time period than the 110- and 145-nm particles. The latter two sizes are in the typical size range of fresh diesel emissions (Charron and Harrison, 2003)". Please explain that.

It was not evident from the text that the first statement referred to LV particles, while the second sentence dealt with total particles. We reformulated one of the sentences and deleted the other sentence to clarify this situation.

**Additional references**

Decesari, S., Facchini, M. C., Matta, E., Lettini, F., Mircea, M., Fuzzi, S., Tagliavini, E., and Putaud, J.-P.: Chemical features and seasonal variation of fine aerosol water-soluble organic compounds in the Po Valey, Italy, Atmos. Environ., 35, 3691–3699, 2001.
Facchini, M. C., Mircea, M., Fuzzi, S., and Charlson, R. J.: Cloud albedo enhancement by surface-active organic solutes in growing droplets, Nature, 401, 257–259, 1999.
Fuzzi, S., Decesari, S., Facchini, M. C., Matta, E., Mircea, M., and Tagliavini, E.: A simplified model of the water soluble organic component of atmospheric aerosol, Geophys. Res. Lett., 20, 4079–4082, 2001.
Salma, I., Ocskay, R., Varga, I., and Maenhaut, W.: Surface tension of atmospheric humic-like substances in connection with relaxation, dilution and solution pH, J. Geophys. Res., 111, D23205, doi:10.1029/2005JD007015, 2006.

Imre Salma
14–02–2018

---

## Author Response (AR2)

**Response to Referee #3**

We thank Referee #3 for this comment. We appreciate and agree with its motivation. Nevertheless, the comparison of hygroscopic and volatility properties of urban-type aerosol particles is above the goals of the present paper due to extension and methodological reasons. We would like to avoid any touch-and-go sections in the MS. To emphasize our primary objectives and the general values of our study, we added the following part to the text:

"The results and conclusion achieved in the present study cover a geographical region in Central Europe, which is much less represented and documented by similar measurements. This can already indicate a general value of our contribution. In addition to that, our findings could in principal be compared to earlier studies performed specifically in urban environments in the word. Nevertheless, there are substantial differences among cities and their associated larger regions, which also means that a perspective comparative analysis should be detailed enough, it is expected to be extended and more importantly, to deal with systematic methods of identifying and quantifying similarities and differences. The present work indicates the need for such an overview work in the future if the number of cities with available corresponding data further increases."

With these sentences, we hope to contribute to initiating a separate, dedicated comparative or review-type study on the corresponding urban results in the word in the future.

Imre Salma
02–03–2018

[revised manuscript text omitted]